# Patterns of ICT Use and Technological Dependence in University Students from Spain and Japan: A Cross-Cultural Analysis

**DOI:** 10.3390/ijerph22050737

**Published:** 2025-05-07

**Authors:** José Antonio Martín Herrero, Ana Victoria Torres García, María Concepción Vega-Hernández, Marcos Iglesias Carrera, Masako Kubo

**Affiliations:** 1Department of Social Psychology and Anthropology, Faculty of Psychology, University of Salamanca, Avda. de la Merced N.º 109-131 C. P, 37005 Salamanca, Spain; 2Department of Personality, Evaluation and Psychological Threatments, Faculty of Psychology, University of Salamanca, Avda. de la Merced N.º 109-131 C. P, 37005 Salamanca, Spain; avit@usal.es; 3Department of Statistics, Higher Polytechnic School of Zamora, University of Salamanca, Av. de Requejo, 33, 49029 Zamora, Spain; mvegahdz@usal.es; 4University School of Labor Relations and Human Resources, University of Salamanca, C. San Torcuato, 43, 49014 Zamora, Spain; marcosiglesias@usal.es; 5Department of Modern Philology, Faculty of Philology, University of Salamanca, Plaza de Anaya, s/n., 37008 Salamanca, Spain; masako@usal.es

**Keywords:** ICT addiction, university students, academic performance, pandemic, COVID-19, cultures and technology

## Abstract

Background: After the end of the COVID-19 pandemic, abusive use of the internet and new information and communication technologies (ICT) among university students was detected. Our research questions were as follows: what has been the impact on the academic performance of university students, and how did the pandemic affect students’ relationship with ICTs? The aim of this research was to explore the use of cell phones and the internet in students from different cultures (Spanish and Japanese) after the pandemic. Methods: This descriptive and exploratory study analysed 206 university students from Spanish and Japanese cultures to understand their perceptions of academic performance after the pandemic, their general use of ICT, and their abusive use of the internet and mobile phones. Instruments included the Internet Overuse Scale (IOS) and the Cell-Phone Overuse Scale (COS), adapted for both Spanish and Japanese populations. Differences between quantitative variables were analyzed using the non-parametric Mann–Whitney U test for independent samples (Spanish and Japanese students or by sex). Contingency tables were created to record and analyse relationships between qualitative variables using the chi-squared test, with statistical significance set at *p* < 0.05. Results: Approximately 29.6% of participants displayed excessive internet use, while 25.2% showed pathological mobile phone use. A strong association was found between high internet and mobile phone usage. Significant cultural and gender differences were observed, with higher problematic use among Japanese students and female participants. Conclusions: Excessive ICT use remains a concern in university settings, with gender and cultural factors playing key roles. These findings highlight the need for targeted digital well-being interventions.

## 1. Introduction

New technologies and social networks satisfy our need to be in contact with people. However, due to the addictive potential of the brain’s response to pleasure, power, and information, the negative part is the appearance of addiction problems due to the abusive use of these tools [1].

Some authors like Brailovskaia [2] and Bai et al. [3] claim that social media (SM) activity can contribute to the development of addictive tendencies, discussing the consequences they have for mental health, and how to prevent them.

According to him Spanish Observatory of Drugs and Addictions, addictions to video games and games are the only two addictions without substances [4]. However, when talking about social media addiction, the only reference is made to “problematic internet use” without including it in diagnostic manuals [5,6].

The abuse of information and communication technologies (ICT) involves the external involvement of human beings with technology related to the internet, computer, mobile phone, video games, etc. [7], characterized by high frequency of use, loss of control by the person, and a dependent relationship with this technology [8]. It is a pattern of behavior characterized by loss of control over the use of the internet. This behavior leads to isolation and neglect of social relationships, academic activities, recreational activities, health, and personal hygiene [9].

Access to these tools is increasing and their use is not without problems. The mobile phone (smartphone) has become the most used technology, and its use is a cause for concern in the field of research and within institutions [10]. The use of this tool is not a problem, but the problematic relationship established with it [11,12] leads to conditioning social relationships when the number of hours of use per day is high or it is used in an uncontrolled manner [13,14].

Given that ICT use is closely shaped by social norms, values, and habits specific to each cultural setting, it is essential to incorporate the cultural dimension when analyzing problematic behaviors associated with it. In this context, comparing samples of university students from culturally distinct environments—such as Spain and Japan—provides an opportunity to examine the extent to which cultural factors may influence patterns of ICT use and misuse. This intercultural perspective not only deepens the theoretical understanding of the phenomenon but also offers empirically grounded insights for the development of more contextually appropriate and culturally responsive strategies in digital education and prevention.

Research in this field has focused on teenagers and young people, detecting the presence of emotional and social behavioral problems related to mobile phone use [15]. Specifically, it has been observed that university students and adolescents are the population groups at greatest risk in this regard [16]. However, we cannot ignore the importance of the mobile phone and the internet for university students, nor the need to use these technologies as tools in the classroom [17,18].

Some authors like Cheng et al. (2023) have discussed the relationship between COVID-19 lockdown stress and the problematic use of social networking sites among quarantined college students in China, following a chain mediation model based on the stressor–strain–outcome framework. They realized that quarantined college students may become concerned about the crucial experiences they potentially miss because of lockdown, and may develop a fear of missing out (FoMO), a condition in which people are highly anxious about missing out on noteworthy experiences that other people are enjoying [19].

Regarding the problems associated with the use of ICT, some studies have pointed out differences regarding gender or differences related to cultural factors [20]. It was found that boys access the internet at earlier ages, while girls access mobile phones earlier, using these to make calls and communicate via WhatsApp, thereby putting them in a situation of greater vulnerability. Gender, therefore, becomes a predictor of excessive use of these tools, especially when it comes to intense or problematic use of the mobile phone or the internet, as the difference between genders increases [21,22,23]. While some studies suggest that being a woman increases the likelihood of developing problematic internet dependence [24], others maintain that it acts as a protective factor against the use of new technologies [25]. Furthermore, other works suggest qualitative gender differences in the use of these technologies, with men having a greater tendency to become involved in games, compared with greater use of social networks by women [26]. Therefore, the impact of gender on ICT use requires further research [20].

Other authors like Abbouyi et al. (2024) have written about a certain connection between the use of the problematic social media use (PSMU) and depression and anxiety symptoms, reporting research that was carried out in the Middle East and North Africa (MENA region) [27].

According to the National Statistics Institute [28] in Spain, 94.5% of the population aged 16 to 74 has used the internet in the past three months, 0.6 percentage points more than in 2021. The gender gap, which in 2017 was 1.8 percentage points, has been disappearing since 2019. The use of the internet is a majority practice among young people; it has been observed that as age increases, its use decreases. The activity most carried out on the internet is instant messaging, such as via WhatsApp.

In response to the COVID-19 pandemic and the fact that the world population had to confine themselves to their homes, as was the case in Spain for almost three months, or limit contacts, as in the case of Japan, which did not require total confinement but did impose important restrictions, the need for communication, work, and academic activity continued through the use of the mobile phone and the internet, which were the most used means of communication.

Touloupis et al. (2023) examined patterns of Facebook use by university students during the period of the public health crisis. Specifically, they investigated the intensity of students’ Facebook use and self-disclosure to unknown online friends, as well as the roles of sense of resilience and loneliness in the manifestation of these behaviors [29].

The objective of the current research was to analyze the use of mobile phones and the internet in Spanish and Japanese university students after the period of the COVID-19 pandemic.

The importance of this study lies in ascertaining whether the pandemic situation, with the subsequent confinement it entailed, may have resulted in an escalation in the levels of ICT use, and whether the population has continued to utilize ICTs with the same (high) frequency as during the confinement situation, even after the conclusion of the pandemic.

The present study is distinct from others that have also examined the consequences of COVID-19 in that it focuses specifically on the use or abuse of ICTs and compares two very different cultures, Spanish and Japanese.

## 2. Materials and Methods

### 2.1. Design

A descriptive, exploratory, cross-sectional, observational study was carried out, where qualitative and quantitative variables associated with the abusive use of ICT and academic performance in university students from Spain and Japan were analyzed after the COVID-19 pandemic.

This study employed a non-probabilistic convenience sampling approach, developed in two phases, in collaboration with two universities: the University of Salamanca (Spain) and the Kyoto University of Foreign Studies (Japan), selected as institutions affiliated with the responsible researchers. The choice of two distinct educational and cultural contexts was made to explore the possible influence of culture on ICT abuse behaviors among university students.

In the first phase, contact was made with the deans’ offices of both universities to request their collaboration and facilitate distribution of the questionnaire. In the second phase, the researchers themselves promoted student participation through the courses they teach. Data collection was conducted using a self-administered online questionnaire, designed on the Google Forms platform and linguistically adapted to the official language of each university (Spanish or Japanese). Access to the questionnaire was open with no additional exclusion criteria; the only inclusion criterion was that participants were enrolled in an official degree program at the time of the study.

Despite efforts to encourage participation, the response rate was limited, probably due to the voluntary nature of the study and the saturation of surveys in the university environment. Ultimately, a total sample of 206 students was obtained.

The questionnaire was developed in 2022, and data collection took place throughout 2024. Invitations to participate were sent via institutional email, and participants had direct contact with the researchers for any inquiries or requests for additional information.

### 2.2. Evaluation Method

In order to collect data, a semi-structured questionnaire prepared for the research was used, consisting of questions to obtain sociodemographic information and details of academic performance and general use of ICT, with scales designed to measure excessive use of the internet and mobile phones, such as the Internet Over-use Scale (IOS) [22] and the Cell-Phone Overuse Scale (COS) [22].

Academic performance was assessed by six items exploring academic outcomes and students’ self-reported perceptions during the COVID-19 pandemic, as follows: “Have you passed all the courses you took during the pandemic?”, “Have your grades and/or academic performance improved since the beginning of the pandemic?”, “Do you think that the changes in teaching methodology during the pandemic period have benefited your academic performance?”, “In the past year, have you noticed an increase in your use of ICT?”, “What type of tool or application do you think you use more than before?”, and “During short breaks from studying, classes, etc., do you use ICT?”.

To assess general patterns of ICT use, 14 items were developed. These items gathered information on frequency, context, and perceived impact of ICT use during and after the COVID-19 pandemic. The questions included the following: “How many hours per day do you use ICTs?”, “In which area do you primarily use ICTs?”, “How frequently do you communicate with your family and/or friends?”, “What communication media do you use most frequently?”, “Has your amount of free time increased as a result of ICT use?”, “Since the beginning of the pandemic, has your online shopping activity increased?”, “If so, what type of products do you typically purchase online?”, “By how much has your average monthly online expenditure increased since the start of the pandemic?”, “During the pandemic, have you engaged in online gambling?”, “If yes, how frequently have you participated in online gambling during the pandemic?”, “If yes, approximately how much money do you gamble per month?”, “Has your use of ICTs changed your eating habits?”, “How many hours of sleep do you get per day since the beginning of the pandemic?”, and “What do you think is the reason for this change?”.

The participants were asked to rate their degree of affectation in relation to the use of the internet and mobile phone, based on the frequency with which they felt, thought, or experienced what the statements indicated, on a Likert-type scale of never (1) to always (6). Examples included the following: “Do you feel worried about whether you have received a call or message and think about it when your phone is off?” and “How often do you make new friends with people connected to Internet?”. Each of these questionnaires (IOS and COS) consisted of 23 questions.

In reference to the psychometric properties of the IOS and COS measurement tools, internal consistency was evaluated using the well-known Cronbach’s alpha and McDonald’s omega. In the Spanish version, the IOS obtained a Cronbach’s alpha coefficient of 0.900 and a McDonald’s omega of 0.901, while the COS reached an alpha of 0.913 and an omega of 0.915. In the Japanese version, the IOS showed an alpha of 0.876 and an omega of 0.868; the COS obtained an alpha of 0.897 and an omega of 0.920.

To calculate the total score for each subject, the responses were recoded as 0 if they were equal to or less than “almost never” or as 1 if they were equal to or greater than “sometimes”; the values obtained ranged from 0 to 23. To classify students with high or low utilization of ICT, the 75th and 25th percentiles were used, respectively [22].

Excessive internet or cell phone use could be considered pathological, so the DSM-5 was used for its assessment [30]. In addition to substance-related disorders, the DSM-5 also includes pathological gambling, which reflects evidence that gambling behaviors activate reward systems similar to those activated by drugs, producing some behavioral symptoms similar to substance use disorders and which may be applicable in the current study setting. The criteria contained in the DSM-5 were used to consider excessive internet and cell phone use as pathological.

### 2.3. Statistical Analysis

A descriptive analysis was carried out considering the main characteristics of the sample used in the study, with reference to the items to used determine perceptions of academic performance after the COVID-19 pandemic through analysis of grades and results obtained at the academic level, as well as general use of ICT, and the abusive use of the internet and mobile phones among university students. For this, frequency tables and measures of central tendency and dispersion were used. In addition, a bar chart was generated to visually illustrate the distribution of the communication media used most among university students, providing a clear representation of participants’ preferences regarding these tools.

Differences between quantitative variables were analyzed using the non-parametric Mann–Whitney U test for independent samples, which is appropriate for comparing two independent groups, such as students from Spanish and Japanese cultural backgrounds or those grouped by sex. This test was selected due to the data not meeting the normality assumptions required for parametric tests.

Relationships between qualitative variables were examined using contingency tables, which were analyzed with the chi-square test. This test is appropriate for assessing independence between categories of qualitative variables.

For the present study, statistical analyses were conducted using a significance level of 0.05. Accordingly, results yielding a *p*-value below this threshold were interpreted as statistically significant. For data analysis, the IBM SPSS Statistics package, version 26.0, was used [31].

### 2.4. Ethics Committee

The research has been approved by the Bioethics Committee of the University of Salamanca, through research protocol number CBE-EP2 1 P-696 28032022.

## 3. Results

The answers obtained corresponded to 146 subjects from the questionnaire in Spanish and 60 in Japanese, with an average age of 21 years (standard error = 0.43) and a greater participation of women (75.7%). In total, 62.1% of the participants in the study were of Spanish nationality, 26.2% were Japanese, and the rest were of other nationalities such as Mexican (5.3%), Chinese (1.9%), or Korean (1.0%), among others, although they lived and studied in Spain or Japan. Regarding types of cohabitation, more than 42% of those surveyed live with parents or relatives, 27.2% lived with roommates, 15% lived alone, almost 8% lived with dorm roommates, and 7.3% lived as part of a couple.

The sample was made up mainly of undergraduate students from Fine Arts, Political Science and Public Management, Criminology, Medicine, Psychology, and Labor Relations and Human Resources degree programs, and 10% were studying for a master’s degree. The reason why there were students from other countries is because master’s students tend to come from different countries. Most of the participants were enrolled in the first year (20%) or third year of undergraduate studies (30%). This study was conducted from 2024, post-pandemic.

### 3.1. Perception of Academic Performance After the COVID-19 Pandemic

If we take academic performance into account, 75.7% indicated that they had passed all the subjects taken during the pandemic, and 53.9% responded that their grades and/or academic performance had increased since the start of the pandemic. Concurrently, 68.9% believed that the modifications in teaching methodology during the pandemic period did not favor their academic performance (1). In the previous year, 87.4% perceived that their time spent using ICT had increased, with the most used tools or applications being Zoom, Microsoft Teams, Google Meet, Google Classroom, Studium, Blackboard, email, Skype, Facetime, LINE, Instagram, Twitter, YouTube, Whatsapp, Tiktok, Google Doc, Office apps (Word, Excel, PowerPoint), graphic applications such as Canva, and other sites such as Turning Point, Quizizz, and Kahoot. Online series and movie platforms included Netflix, among others, and online sales platforms included Aliexpress. Furthermore, 26.2% confessed that they always used ICT in their short study breaks, classes, etc., and 71.8% used it sometimes or almost always.

### 3.2. General Use of Information and Communication Technologies

In relation to the daily use of ICT, less than 15% of students used these resources for less than 3 h, 26.2% indicated that the time they spent was 3 to 5 h, more than 46% used them for 5 to 10 h, 5.3% did so from 10 to 15 h, and almost 3% said they used them for too many hours or even from the moment they got up until they went to bed.

Referring to the areas where they used ICT the most, in this sample of students, 45.1% of them used ICT for their studies and 2.9% did so in the workplace; 21.8% used ICT to communicate through calls or chats; 23.3% used ICT to watch series or movies, read, listen to music, etc., and 2.9% to play online games. Almost 80% of students said that they communicated with their family and/or friends every day, and only 2% indicated that they barely did so or did so once a week.

Students used different means to communicate, with the most used means of communication being Whatsapp (71.4%), Instagram (58.7%) and traditional telephone calls (46.1%), followed by Line (24.8%) and Facebook (14.6%). The least used means of communication included Messenger (3.9%) and Skype (5.8%), in part because use of the latter has fallen as other forms of communication such as Telegram have appeared (3.9%). Other means of communication they used included Discord, Zoom, Meet, Facetime, Twitter, and TikTok (See Figure 1).

In this study, 64.6% of the research participants indicated that their free time had not increased because of the use of ICT. Regarding consumption, 60.2% indicated that since the beginning of the pandemic their online purchases had increased, with this consumption being related to food, clothing, books, games, electronic devices, accessories, etc. More than half saw their average monthly online spending increase after the start of the pandemic by between EUR 20 and 50, 15% increased their online spending by between EUR 51 and 100, and almost 4% by more than EUR 100 (Table 1). Regarding online gambling, only 1% of those surveyed made online bets during the pandemic, almost every day or once every two weeks, with the amount invested in bets being about EUR 20 per month.

Table 1 shows that 32% indicated that the use of ICT had changed their eating habits; 13.1% indicated that they were eating more healthily, 12.1% that they ate more fast food or takeaways, and 6.8% more snacks, sweets, soft drinks, etc. Furthermore, since the beginning of the pandemic, only 59.7% had slept between 7 and 8 h a day, 23.3% slept between 5 and 6, 14.1% more than 8 h, and almost 3% slept less than 5 h. Among the reasons why they believed their sleeping hours had changed, participants referred to aspects related to the pandemic situation itself, such as having to spend more time at home due to the curfew, demotivation, and even anhedonia and hopelessness. Through not having expectations for the future, feeling listless, and due to the need to occupy their time, they slept more.

### 3.3. Abusive Use of the Internet in Spanish and Japanese Universities

Table 2 reflects the percentages of responses regarding normal or abusive (pathological) use of the internet revealed through the application of the IOS, considering the frequency with which respondents felt, experienced, or thought what the statements on that scale indicated, with good internal consistency (α = 0.893 and ω = 0.894).

Most of the university students say they sometimes felt worried about what happens on the internet, they thought about it when they were not connected and abandoned activities or tasks to spend more time connected to the internet (items 1 and 3). Many of them, 33%, reported that they sometimes connected to the internet to escape their problems (item 8). Most also indicated that sometimes they stay connected for longer than they initially thought, lost track of time when they were online, or had felt guilty for investing too much time in their connections (items 19, 21, and 22).

In total, 29.6% of the sample made excessive use of the internet, and these could be considered pathological users. Comparing the students who responded to the questionnaire in Spanish and in Japanese, statistically significant differences were observed only in the DSM-5 criterion 8 score [30] which refers to the subject having endangered or lost an important relationship, job, or academic or professional career due to abusive use of the internet (Mann–Whitney U = 5231.5; *p*-value = 0.023). In the other results, there was no evidence to say that students from the two cultures were different. We also observed statistically significant differences by sex in the scores associated with tolerance according to criterion 1 of the DSM-5 [30], which refers to the need to invest increasing amounts of time on the internet to achieve the desired level of arousal (Mann–Whitney U = 3133; *p*-value = 0.024).

Table 3 shows that among the students who responded to the questionnaire in Japanese, 56.8% used the internet excessively, while this was the case for less than half of those who responded in Spanish (42.9%). Among female students, half used the internet excessively, but among men the percentage was lower (40%).

### 3.4. Abusive Use of Mobile Phones in Spanish and Japanese Universities

Table 4 shows the percentages of responses found regarding normal or abusive (pathological) use of mobile phones, according to the COS. The Cronbach’s alpha obtained in the study sample for this scale was α = 0.907 and the McDonald’s omega was ω = 0.910, which indicates excellent internal consistency.

Most of the sampled students sometimes felt worried about whether they have received a call or message and thought about it when their phone was off (item 1); they also reported that they used their cell phones to escape from their problems (item 8). Most of them used their mobile phone for longer than they initially intended, lost track of time when using their mobile phones, and had felt guilty for spending too much time using their mobile phone (items 19, 21, and 22).

The results reveal that 25.2% of the sample made excessive use of their mobile phones; these could be considered pathological mobile phone users. Statistically significant differences considering sex were observed only in the scores forDSM-5 criterion 1 [30], which refers to the need to invest increasing amounts of time on the internet to achieve the desired level of arousal (Mann–Whitney U = 3047.5; *p*-value = 0.013). The other survey results included no evidence of sex-based differences.

According to the results in Table 5, among the students who responded to the questionnaire in Japanese, 51.4% used mobile phones excessively, and among those who responded in Spanish, 45.5% used mobile phones excessively. Among male students, 34.6% uses mobile phones excessively; however, in women, the percentage was higher, including more than half of respondents (54.4%).

Finally, Table 6 presents results for the students classified with light or severe use of both the internet (IOS) and mobile phones (COS). A significant association was observed between the two abusive use classification variables (*p*-value < 0.001); 48.2% of the sample made excessive use of the internet and mobile phones. There were no university students who reported using the internet lightly and the mobile phone excessively. Of those who used the internet excessively, 87.2% abused their mobile phones, and among those who used mobile phones excessively, all of them used the internet excessively.

## 4. Discussion

Addiction to the internet and, by extension, to all technologies related to information and communication, is described as an abusive use that can cause interference or changes in life. The period of the COVID-19 pandemic was not managed in the same way in the Spanish and Japanese populations. In Spain, there was total confinement for almost three months and Japan maintained restricted activity, without imposing total confinement.

The COVID-19 pandemic acted as a catalyst for students’ relationships with ICT, as the increase in virtuality and dependence on technology was exacerbated during this period. This phenomenon raises questions about how the health crisis impacted addiction to new technologies and the emotional well-being of university students.

An investigation carried out on internet addiction in a sample of young Malaysians detected a high risk of internet addiction among young people between 18 and 25 years old and especially among university students [32]. In our research, we observed, that the use of the internet caused changes, in the same way as in the Malaysian population, with mild dependence observed in 43.2% of university students, moderate dependence in 51.0%, and severe dependence in 4.9%, and only 1% made appropriate use of the internet.

If we consider the time that university students dedicate to new technologies, the results of our research are in line with other research, despite the temporal differences of the various studies. The use of new technologies is very high [33]; it was reported that the average daily use of new technology was 6.5 h, and the time dedicated to the new technologies was 4.83 h [3]. As possible explanations for these differences, we can suggest the ages of the subjects, the average age being higher, as well as the ways of relating to and using the new technologies (they were used more outside the home, in different forms of social interaction, or as later arrivals to the digital world). On the other hand, in our research, it is important to highlight the use given to these tools by university students; the highest percentage was used for studying, followed by watching series and movies, reading, listening to music, and as a means of communication, including chats. In contrast, a very small number used them for online games or as tools for work. In research carried out [34] regarding the frequency of use of the internet, video games, mobile phones and television, it was highlighted that almost all students connected to the internet every day (97.8% of the students at the University of Granada and 99.7% of those at the University of Almería). The use of mobile phones for leisure was also reported; 92.59% of Granada university students used their device every day compared with 69.44% of Almeria university students. With respect to daily television use, students from the Universities of Granada and Almería reported very similar percentage rates, 56.61% and 60.29%, respectively, and these results are in line with our research.

It has been observed that the abusive use of ICT`s can have a significant impact on the academic performance of university students, as has been pointed out in previous research. Addiction to new technologies can lead to problems such as absenteeism, exam failure, and even expulsion, suggesting the need to address this issue comprehensively in educational settings.

In relation to the level of dependency and concern on the part of the students about what is happening on the internet, according to our results, almost 30% of the sample used the internet excessively and could be considered pathological internet users. These results contrast with those obtained in other investigations in which the risk of internet addiction was low [35,36,37].

With respect to the use of technologies and sex, in our research, according to the results that were obtained via the IOS questionnaire, mild and severe levels of internet use occur more in women; these results reveal an opposite trend to other research regarding internet addiction. Orozco-Calderon [38] found that in the majority of the subjects evaluated, groups of men with medium and severe addiction had higher scores than women, along the same lines as other research that relates gender with internet addiction symptomatology and indicates a higher prevalence in men, children, and adolescents compared to women [39,40,41,42]. Lam and Peng [37] point out that young men spend more time on the internet in individual and team activities, games, and on adult sites, and that young women use social networks more.

We highlight the importance of considering gender differences in internet use, as previous studies have shown that there are disparities in the way men and women use ICTs. These findings underscore the need to design specific strategies that address the different ways in which genders interact with technology.

Our study highlights the need to continue researching the use and abuse of ICTs in educational contexts, considering factors such as academic performance, gender differences and the context of the pandemic. These findings can be fundamental for the design of interventions and policies that promote the healthy and balanced use of information and communication technologies in the student population.

Our results agree with Jenaro et al. (2007), which validates the scales used in this study [22].

To support the link between excessive mobile phone use and mental health, we refer to the study by Hashemi et al. (2022), which examines the connection between excessive mobile phone use and symptoms of depression, anxiety, and stress in university students (BioMed Central) [43].

Regarding the study of the DSM-5 criteria for internet addiction, we acknowledge the work of Pontes and Griffiths (2015), who developed and validated a psychometric scale based on these criteria for internet gaming disorder [44]. Our results are like those of these authors.

In relation to the validation of scales in diverse cultural contexts, we consider the study by Ching et al. (2015), which validated a Malaysian version of the Smartphone Addiction Scale [45]. This study highlights the importance of adapting and validating instruments in different cultural contexts.

## 5. Conclusions

The COVID-19 pandemic has increased the dependence on ICTs in the lives of university students, raising the need to proactively address the potential negative effects of this increased exposure to technology. Negative impacts were observed on the academic performance of university students and their habits, suggesting the need to implement prevention and awareness strategies to encourage healthy use of technology.Significant increases have been reported in ICT addiction among young adults, with almost 30% being pathological internet users and 25.2% considered pathological mobile phone users.There is a clear association between excessive use of the internet and abusive use of mobile phones, with women being the most abusive users. Significant differences are evident in the use of ICT between men and women, which highlights the importance of considering gender approaches when designing interventions related to the use of ICTs in educational environments.Despite the adaptations in teaching during the pandemic, most participants (68.9%) did not perceive improvements in their academic performance, which suggests the need to review and adjust educational strategies in virtual environments.There are significant differences in the use of the internet and mobile phones between Spanish and Japanese students, which highlights the influence of cultural factors on technological dependence. These disparities can have an impact on the academic performance and mental health of young people.Increased use of ICTs during the pandemic has led to increased problematic use of the internet and mobile phones. These findings suggest the need to address ICT addiction and its potential consequences on mental health and academic performance.

In summary, internet addiction and the abusive use of technologies are relevant phenomena that affect university students, with gender and cultural differences in their manifestation. It is essential to address these issues to promote healthy use of technology and prevent potential negative impacts on young people’s academic and personal lives.

We believe it is necessary to continue researching the use and abuse of ICTs, considering cultural differences, to better understand how these technologies impact the academic and personal lives of university students and thus, the implications of the use of ICTs in terms of effects on the emotional, social, and academic well-being of university students, as well as to develop effective strategies for the prevention and treatment of possible technological addictions.

### 5.1. Study Biases and Limitations

Among the limitations of the study, one of which is the sample size and the difference between the Spanish and Japanese populations, it is indicated in the manuscript that the fact of belonging to two universities from different countries and cultures could bias the interpretation of the comparative statistical results since we cannot consider the weight of the independent variable “culture”. Nevertheless, we suggest as a prospective to continue with research in relation to the use and abuse of information and communication technologies in both cultures as well as to go deeper into cultural differences.

### 5.2. Implications of This Study

Governmental and educational authorities are now beginning to give importance to behavioral addictions, including ICTs. It is appropriate that government agencies (medical planning departments and other related institutions) begin to review whether the origin of the spike in the prevalence of ICT addiction stems from the period of confinement produced by COVID-19.

## Figures and Tables

**Figure 1 ijerph-22-00737-f001:**
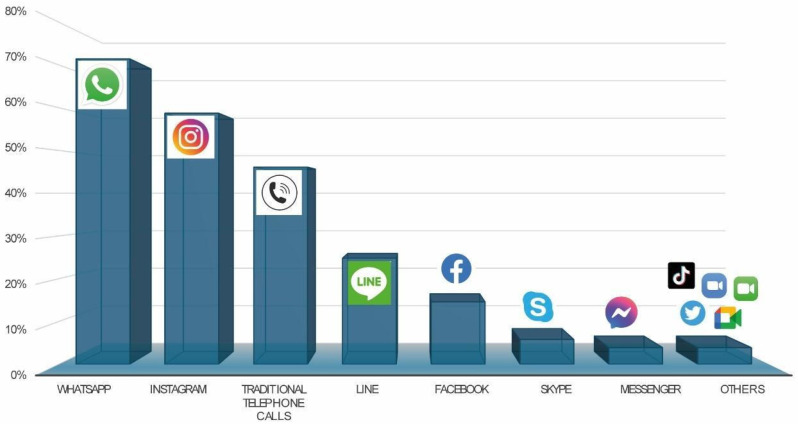
Media most used by university students.

**Table 1 ijerph-22-00737-t001:** Distribution of changes in the habits of university students after the COVID-19 pandemic.

	n	%
**How much has your average monthly online spending increased since the start of the pandemic?**		
Between EUR 20 and EUR 50 (or the equivalent in your local currency)	118	57.3%
Between EUR 51 and EUR 100 (or the equivalent in your local currency)	31	15.0%
More than EUR 100 (or the equivalent in your local currency)	8	3.9%
I do not make online purchases	49	23.8%
**Has the use of ICT changed your eating habits?**		
Yes, I eat more fast food or takeout	25	12.1%
Yes, I eat more snacks, sweets, soft drinks, etc.	14	6.8%
Yes, I eat healthier	27	13.1%
No, I have the same eating habits	140	68.0%
**How many hours have you slept a day since the start of the pandemic?**		
More than 8 h	29	14.1%
Between 7 and 8 h	123	59.7%
Between 5 and 6 h	48	23.3%
Less than 5 h	6	2.9%

**Table 2 ijerph-22-00737-t002:** Responses to excessive internet use scale (IOS).

	Never		Hardly Ever		Sometimes		Often		Almost Always		Always	
IOS	n	%	n	%	n	%	N	%	n	%	n	%
**item 1**	32	15.5%	63	30.6%	76	36.9%	18	8.7%	11	5.3%	6	2.9%
**item 2**	56	27.2%	82	39.8%	46	22.3%	14	6.8%	7	3.4%	1	0.5%
**item 3**	16	7.8%	67	32.5%	71	34.5%	38	18.4%	9	4.4%	5	2.4%
**item 4**	89	43.2%	56	27.2%	42	20.4%	9	4.4%	5	2.4%	5	2.4%
**item 5**	164	79.6%	22	10.7%	14	6.8%	2	1.0%	2	1.0%	2	1.0%
**item 6**	63	30.6%	61	29.6%	49	23.8%	19	9.2%	8	3.9%	6	2.9%
**item 7**	148	71.8%	42	20.4%	12	5.8%	4	1.9%	-	-	-	-
**item 8**	52	25.2%	49	23.8%	68	33.0%	17	8.3%	15	7.3%	5	2.4%
**item 9**	80	38.8%	40	19.4%	54	26.2%	18	8.7%	9	4.4%	5	2.4%
**item 10**	64	31.1%	57	27.7%	53	25.7%	19	9.2%	9	4.4%	4	1.9%
**item 11**	60	29.1%	62	30.1%	55	26.7%	12	5.8%	9	4.4%	8	3.9%
**item 12**	120	58.3%	64	31.1%	17	8.3%	2	1.0%	2	1.0%	1	0.5%
**item 13**	170	82.5%	26	12.6%	9	4.4%	-	-	1	0.5%	-	-
**item 14**	116	56.3%	59	28.6%	27	13.1%	2	1.0%	2	1.0%	-	-
**item 15**	118	57.3%	46	22.3%	34	16.5%	6	2.9%	2	1.0%	-	-
**item 16**	67	32.5%	47	22.8%	59	28.6%	16	7.8%	12	5.8%	5	2.4%
**item 17**	153	74.3%	40	19.4%	10	4.9%	1	0.5%	1	0.5%	1	0.5%
**item 18**	147	71.4%	40	19.4%	14	6.8%	4	1.9%	1	0.5%	-	-
**item 19**	19	9.2%	17	8.3%	73	35.4%	42	20.4%	41	19.9%	14	6.8%
**item 20**	79	38.3%	53	25.7%	51	24.8%	18	8.7%	5	2.4%	-	-
**item 21**	28	13.6%	37	18.0%	59	28.6%	41	19.9%	29	14.1%	12	5.8%
**item 22**	36	17.5%	39	18.9%	66	32.0%	30	14.6%	23	11.2%	12	5.8%
**item 23**	82	39.8%	56	27.2%	50	24.3%	9	4.4%	7	3.4%	2	1.0%

**Table 3 ijerph-22-00737-t003:** Contingency tables of the classification of ICT use (light/heavy) according to the IOS by culture (questionnaire in Spanish/Japanese) and sex.

			QUESTIONNAIRE		SEX		Total
			Spanish	Japanese	Women	Man	
**IOS**	Light	**n**	48	19	46	21	67
		**% within IOS**	71.6%	28.4%	68.7%	31.3%	100.0%
		**% within questionnaire or sex**	57.1%	43.2%	49.5%	60.0%	52.3%
	Heavy	**n**	36	25	47	14	61
		**% within IOS**	59.0%	41.0%	77.0%	23.0%	100.0%
		**% within questionnaire or sex**	42.9%	56.8%	50.5%	40.0%	47.7%
Total		**n**	84	44	93	35	128
		**% within IOS**	65.6%	34.4%	72.7%	27.3%	100.0%
		**% within questionnaire or sex**	100.0%	100.0%	100.0%	100.0%	100.0%

**Table 4 ijerph-22-00737-t004:** Mobile phone excessive use scale (COS) responses.

	Never		Hardly Ever		Sometimes		Often		Almost Always		Always	
COS	n	%	n	%	n	%	n	%	n	%	n	%
**item 1**	42	20.4%	44	21.4%	72	35.0%	25	12.1%	16	7.8%	7	3.4%
**item 2**	71	34.5%	77	37.4%	47	22.8%	9	4.4%	1	0.5%	1	0.5%
**item 3**	37	18.0%	77	37.4%	62	30.1%	22	10.7%	4	1.9%	4	1.9%
**item 4**	106	51.5%	57	27.7%	32	15.5%	7	3.4%	1	0.5%	3	1.5%
**item 5**	159	77.2%	34	16.5%	8	3.9%	3	1.5%	2	1.0%	-	-
**item 6**	86	41.7%	52	25.2%	45	21.8%	10	4.9%	7	3.4%	6	2.9%
**item 7**	148	71.8%	42	20.4%	12	5.8%	3	1.5%	1	0.5%	-	-
**item 8**	65	31.6%	43	20.9%	66	32.0%	17	8.3%	10	4.9%	5	2.4%
**item 9**	90	43.7%	49	23.8%	37	18.0%	17	8.3%	10	4.9%	3	1.5%
**item 10**	69	33.5%	60	29.1%	40	19.4%	24	11.7%	7	3.4%	6	2.9%
**item 11**	78	37.9%	64	31.1%	40	19.4%	14	6.8%	4	1.9%	6	2.9%
**item 12**	126	61.2%	53	25.7%	24	11.7%	2	1.0%	1	0.5%	-	-
**item 13**	169	82.0%	27	13.1%	8	3.9%	2	1.0%	-	-	-	-
**item 14**	107	51.9%	52	25.2%	32	15.5%	11	5.3%	3	1.5%	1	0.5%
**item 15**	125	60.7%	47	22.8%	24	11.7%	8	3.9%	1	0.5%	1	0.5%
**item 16**	75	36.4%	61	29.6%	40	19.4%	20	9.7%	9	4.4%	1	0.5%
**item 17**	169	82.0%	26	12.6%	8	3.9%	1	0.5%	1	0.5%	1	0.5%
**item 18**	135	65.5%	44	21.4%	18	8.7%	5	2.4%	3	1.5%	1	0.5%
**item 19**	27	13.1%	24	11.7%	63	30.6%	39	18.9%	34	16.5%	19	9.2%
**item 20**	73	35.4%	54	26.2%	57	27.7%	18	8.7%	3	1.5%	1	0.5%
**item 21**	37	18.0%	34	16.5%	55	26.7%	44	21.4%	22	10.7%	14	6.8%
**item 22**	37	18.0%	46	22.3%	65	31.6%	30	14.6%	17	8.3%	11	5.3%
**item 23**	87	42.2%	49	23.8%	48	23.3%	13	6.3%	6	2.9%	3	1.5%

**Table 5 ijerph-22-00737-t005:** Contingency tables of the classification of ICT use (light/heavy) according to the COS by culture (questionnaire in Spanish/Japanese) and sex.

			QUESTIONNAIRE		SEX		Total
			Spanish	Japanese	Women	Man	
**IOS**	Light	**N**	35	18	36	17	53
		**% within IOS**	66.0%	34.0%	67.9%	32.1%	100.0%
		**% within questionnaire or sex**	48.6%	54.5%	45.6%	65.4%	50.5%
	Heavy	**N**	37	15	43	9	52
		**% within IOS**	71.2%	28.8%	82.7%	17.3%	100.0%
		**% within questionnaire or sex**	51.4%	45.5%	54.4%	34.6%	49.5%
Total		**N**	72	33	79	26	105
		**% within IOS**	68.6%	31.4%	75.2%	24.8%	100.0%
		**% within questionnaire or sex**	100.0%	100.0%	100.0%	100.0%	100.0%

**Table 6 ijerph-22-00737-t006:** Contingency table between the classification of the use of ICT via IOS and COS.

			COS CLASSIFICATION		Total
			Light	Heavy	
**IOS CLASSIFICATION**	Light	**n**	38	0	38
		**% within IOS**	100.0%	0.0%	100.0%
		**% within COS**	86.4%	0.0%	44.7%
		**% of the total**	44.7%	0.0%	44.7%
	Heavy	**n**	6	41	47
		**% within IOS**	12.8%	87.2%	100.0%
		**% within COS**	13.6%	100.0%	55.3%
		**% of the total**	7.1%	48.2%	55.3%
Total		**n**	79	26	105
		**% within IOS**	51.8%	48.2%	100.0%
		**% within COS**	100.0%	100.0%	100.0%
		**% of the total**	51.8%	48.2%	100.0%

Chi-square = 64.038; df = 1; *p*-value < 0.001.

## Data Availability

The original contributions presented in this study are included in the article. Further inquiries can be directed to the corresponding author.

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
