# Peer review of "Patterns of ICT Use and Technological Dependence in University Students from Spain and Japan: A Cross-Cultural Analysis"

_ijerph, 2025, doi:10.3390/ijerph22050737_

Round 1
Reviewer 1 Report
Comments and Suggestions for Authors
This study, whose main objective is to analyse the use of mobile phones and the Internet among Spanish and Japanese university students after the COVID-19 pandemic period, essentially addresses an important issue. However, the following recommendations can be made to the authors regarding some corrections that should be made to the content of the study:
The abstract needs to be rewritten and the term "summary" should be replaced by "abstract".
It is not clear why the following two questions are included in the abstract: “What
has been the impact on the academic performance of university students? How did the pandemic affect the relationship of students with ICT?. If these are research questions, it should be clearly stated that they are research questions.
The spelling of covid-19 must be in capital letters. It appears that the authors write OCVID-19 in lower case.
The introduction of the study should be reconsidered and the importance of this study and how it differs from other studies should be clearly stated. In addition, the international literature on both COVID-19 and ICT use is very rich and the literature section of a research on this topic should be more comprehensive.
The methodology section requires more detailed explanations. For instance, the researchers utilized the Mann-Whitney U test, and it is important to clarify why this non-parametric analysis was selected.
Additionally, the tables in the study should be reorganized. As they stand, the tables are difficult to read and understand. I recommend that the authors make them more concise.
Furthermore, the study should address what specific recommendations can be made to policymakers and government officials based on the findings. Currently, the implications for government and education authorities are not addressed in the research.
Incorporating these recommendations will improve the clarity and quality of the study. I recommend that the authors revise the paper in accordance with these recommendations.
Author Response
THE SUGGESTIONS RECEIVED HAVE BEEN CARRIED OUT. THE REVIEWER CAN SEE THEM IN BLUE COLOR IN THE ATTACHED TEXT-MANUSCRIPT-SUBMISSION.
Corrected the duplication of the abstract from the original manuscript.
Emphasized that these two questions are basic to our article and constitute our research questions.
Corrected and capitalized COVID-19.
The bibliography has been corrected and updated with some publications of interest related to the topic of study.
Added in the introduction the importance of this study and how it differs from other studies.
Explained why the Mann-Whitney U test was selected.
Reorganized and corrected the study tables to make them more visible and easier for the reader to interpret.
We have addressed what specific recommendations can be made to policy makers and government officials based on the findings.
We sincerely appreciate your comment and suggestions, which will undoubtedly help us improve the quality of the manuscript.
Regarding the use of the Mann-Whitney U test, we will include a more detailed explanation of the rationale behind selecting this non-parametric analysis. The decision was based on the fact that the variables under analysis did not follow a normal distribution, making the use of parametric tests inappropriate. This non-parametric test was therefore chosen to compare differences between independent groups with respect to qualitative variables, ensuring the validity of the results without assuming normality in the data.
As for the tables presented in the study, we acknowledge your recommendation and will proceed to revise them accordingly.
Reviewer 2 Report
Comments and Suggestions for Authors
Thank you for giving me an opportunity to review this article regarding the influence of ICT use on Spanish and Japanese college students after the COVID-19 Pandemic. The authors lay out the arguments clearly, while the contributions seem to be mainly on the descriptive level. Upon closer examination, it is evident that this paper suffers from significant deficiencies in its theoretical foundation and methodology, which undermines the scientific validity of its conclusions. Specifically, the paper exhibits notable issues in at least five areas:
- The study examined 146 samples from Spain and 60 samples from Japan. However, the authors failed to provide essential sampling information (such as survey timing and criteria for school selection). Clearly, these samples were not obtained through random sampling procedures, rendering any statistical results derived from them unreliable.
- The authors aim to explore the differential impact of ICT use on college students across two cultural contexts. Yet, the rationale behind conducting such a comparative analysis is not adequately explained, leading to fundamental flaws in the study design.
- From a normative perspective, the paper lacks a clear problem awareness and does not include a dedicated literature review section. The impact of ICT use on college students is only briefly mentioned, with the authors suggesting it may affect academic performance. A more precise definition of this impact is necessary.
- Methodologically, the correlation analysis employed by the authors does not account for confounding factors. It is recommended that the authors utilize multiple regression analysis or other appropriate methods to re-examine the data.
- Based on the findings, the authors should provide additional practical policy recommendations.
In summary, although the topic of this manuscript is attractive, the paper requires a Major Revision to beef up its theoretical framing, methodology and explanation of results etc.
Author Response
The reviewer will find the modifications in the text in green color, following your suggestions.
We appreciate your observation and the opportunity to clarify the sampling procedure used in the study. We confirm that the samples were not obtained through simple random sampling, but rather through non-probabilistic convenience sampling, a common strategy in exploratory studies of an educational and sociocultural nature, especially when access to the population is conditioned by participant availability or institutional limitations. We are fully aware of the limitations inherent in the sampling method used, particularly regarding the generalization of the results. For this reason, we have framed the analysis as exploratory and have interpreted it with the necessary methodological caution. We will provide more detailed information about this in the methodology section and discuss its implications in the study’s limitations section.
- The study examined 146 samples from Spain and 60 samples from Japan. However, the authors failed to provide essential sampling information (such as survey timing and criteria for school selection). Clearly, these samples were not obtained through random sampling procedures, rendering any statistical results derived from them unreliable.
We have discussed how the sample was recruited. It was done randomly since the data were obtained through questionnaires sent on-line through the Google Forms platform.
- The authors aim to explore the differential impact of ICT use on college students across two cultural contexts. Yet, the rationale behind conducting such a comparative analysis is not adequately explained, leading to fundamental flaws in the study design.
The reason for using two culturally different samples is explained in the introduction. We tried to find out how much weight the independent variable “culture” has on the dependent variable ICT abuse behavior.
We appreciate your comment. The decision to study two universities in different cultural contexts (Spain and Japan) is driven by the interest in exploring the impact of culture on ICT abuse among university students. The literature suggests that cultural norms may influence behaviors toward ICT, but few studies directly compare these behaviors in such distinct contexts. This comparative approach aims to fill that gap, providing an intercultural perspective that could reveal significant differences in ICT abuse behaviors. We will include a more detailed explanation of this justification in the introduction and methodology sections to better contextualize the approach within the theoretical framework.
- From a normative perspective, the paper lacks a clear problem awareness and does not include a dedicated literature review section. The impact of ICT use on college students is only briefly mentioned, with the authors suggesting it may affect academic performance. A more precise definition of this impact is necessary.
We appreciate your comment and understand that a more detailed review of the literature and a clearer definition of the impact of ICT on university students are essential aspects to strengthen the theoretical framework of the study.
In response to your observation, we will include a section dedicated to the literature review, where we will thoroughly address previous studies on the impact of ICT on university students, particularly in relation to academic performance, time management, and behaviors associated with ICT abuse. We will also clarify how the use of ICT can influence these aspects both positively and negatively, providing a more complete and precise view of the issue. Additionally, we will revise the section where the impact of ICT is mentioned, adding a clearer and more detailed definition of the potential impact these technologies have on university students, both academically and behaviorally.
- Methodologically, the correlation analysis employed by the authors does not account for confounding factors. It is recommended that the authors utilize multiple regression analysis or other appropriate methods to re-examine the data.
We appreciate your comment and the opportunity to clarify the methodological approach used in the study. We would like to point out that a correlation analysis was not conducted, as most of the variables analyzed are qualitative (categorical) in nature. For this reason, we opted to use contingency table analysis, which we consider more appropriate for exploring relationships between variables in the context of this study. Nevertheless, we understand the importance of considering potential confounding factors, and depending on the nature of the data, we are evaluating the possibility of complementing the analysis with additional statistical models in the future.
We will include this clarification in the statistical analysis section to explicitly reflect the rationale behind the choice of analytical methods. - Based on the findings, the authors should provide additional practical policy recommendations.
A section Implications of the study has been added. Currently, governmental and educational authorities are beginning to give importance to behavioral addictions, including ICTs. It is appropriate that government agencies (drug plans and other institutions involved) begin to review whether the origin of the spike in the prevalence of ICT addiction stems from the period of confinement produced by COVID-19.
Reviewer 3 Report
Comments and Suggestions for Authors
Dear author, the article deserves to be reviewed in detail: it is important to be clear about the objective of the study and that it is consistent with the title, the research questions, as well as the results and the discussion of these. It is important that most of your references are not older than 5 years. Specific observations are noted in the corresponding section.
- Introduction
Dear author, I recommend that you be more specific in the introduction, as well as the statement of the problem and the justification of this, it is presented in a very general way the ICT, and it exposes little about the use of mobile phones and the Internet, this being the main objective of the study. Likewise, there must be an agreement between what is presented in the summary, the introduction, the results, the discussion and conclusion
Title; INFLUENCE OF THE USE OF ICT IN UNIVERSITY STUDENTS FROM DIFFERENT CULTURES AFTER THE
COVID-19 PANDEMIC.
Research questions: What has been the impact on the academic performance of university students? How did the pandemic affect the relationship of students with ICT?
Objective of the study: explore the use of mobile phones and the Internet in students from different cultures (Spanish and Japanese) after the pandemic.
- Material and methods
In the section on measurement instruments, have the instruments been validated in different countries (Spain and Japan), who validated them, what was the value of Cronbach's Alpha and McDonald's Omega?. the Validation in different countries is very importante to consider cultural differences and to be able to compare the presence of this problem.
Regarding the section “Excessive use of the Internet or mobile phone could be considered pathological, so the DSM-V was used for its evaluation.25 In addition to substance-related disorders, the DSM-V also includes pathological gambling, which reflects evidence that gambling behaviors activate reward systems similar to those activated by drugs, producing some behavioral symptoms similar to substance-related disorders. substance consumption and that may be applicable on this occasion”. You say that you used the criteria contained in the DSM-5 to consider excessive use of the Internet or mobile phone as pathological?
In the statistical analysis, the perception of academic performance is discussed, but the description of the instruments does not describe how this variable will be evaluated, and it is not stated in the introduction either.
The results show classifications of the results, for example: pathological and excessive use of the Internet and video games, but no cut-off points are established in the corresponding section.
What were the inclusion, exclusion and elimination criteria?
Results
In the abstract and in the objective, it talks about students from different cultures (Spanish and Japanese) and in the results, it points out university students from other countries.
Also, results are presented that talk about activities that students did during the pandemic, so the question arises: is it a study carried out during the pandemic or after the pandemic?
It is important to present the results according to the objectives of the study.
This article is suggested doi: 10.1007/s10654-016-0149-3
- Discussion
Addiction and excessive use are presented in the discussion part, it would be convenient to establish these differences from the methodological part and address them in the introduction.
To make statistical inferences, it is suggested to read this article doi: 10.1007/s10654-016-0149-3
Establish an order in the presentation of the results and the discussion according to the main objectives of the study.
There are many limitations of the study and these are not stated
Author Response
THE SUGGESTIONS RECEIVED HAVE BEEN CARRIED OUT. THE REVIEWER CAN SEE THEM IN BROWN COLOR IN THE ATTACHED TEXT-MANUSCRIPT-SUBMISSION.
We sincerely appreciate your comments.
Indeed, we are aware of the importance of validating measurement instruments in different cultural contexts, especially when the aim is to conduct cross-cultural comparisons. In our study, the questionnaires were linguistically adapted into Spanish and Japanese by native-speaking researchers with university teaching experience and an understanding of the respective cultural contexts. Although a full formal cross-cultural validation was not conducted, a semantic and content review was carried out to ensure conceptual equivalence of the items. Regarding internal reliability, Cronbach’s alpha and McDonald’s omega coefficients were calculated. In the overall study sample, the IOS scale yielded an alpha of 0.893 and an omega of 0.894, while the COS scale showed an alpha of 0.907 and an omega of 0.910. In the Spanish version, the IOS scale reported a Cronbach’s alpha of 0.900 and McDonald’s omega of 0.901, and the COS scale obtained an alpha of 0.913 and an omega of 0.915. In the Japanese version, the IOS scale presented an alpha of 0.876 and an omega of 0.868, whereas the COS scale yielded an alpha of 0.897 and an omega of 0.920. These values will be included to facilitate a clearer assessment of the internal consistency of the instruments used.
We acknowledge, however, the methodological limitation of not having conducted a full cross-cultural validation, and this will be noted in the limitations section, suggesting that future research should explore this aspect in greater depth.
Regarding the question about the DSM-5, our intention was to provide a theoretical framework for understanding problematic ICT use, referring to the fact that the DSM-5 recognizes certain behavioral addictions—such as gambling disorder—that share reward mechanisms with excessive use of digital technologies. In this sense, it may serve as a useful conceptual reference. However, no clinical diagnosis was made in our study, as the instruments employed were self-report questionnaires designed to assess problematic ICT behaviors.
We appreciate your observation regarding the description of the instrument, as it allows us to clarify and improve the presentation of the measurement tools used.
Additionally, to classify students with excessive or light ICT use, the 75th and 25th percentiles were used, respectively. This is now specified in the evaluation method section.
We sincerely appreciate your observation, which allows us to clarify several important aspects of the study.
First, regarding the mention of university students from “other countries,” this was a wording error. The study sample consists exclusively of students enrolled at one Spanish and one Japanese university. However, in today’s globalized world, it is possible that these students were born in different countries, even though they currently reside and study in Spain or Japan. We considered it relevant to collect this information. We will carefully revise the text to avoid any confusion on this point.
Second, in relation to the timing of the study, data collection was conducted after the acute phase of the COVID-19 pandemic. However, some of the questionnaire items referred to experiences during the pandemic in order to assess potential behavioral changes resulting from that period. We acknowledge that this may lead to confusion if not presented clearly, and we will therefore include an explicit clarification in the methodology section.
We fully agree on the importance of aligning the results with the stated objectives and appreciate the bibliographic reference provided, which we have reviewed to improve both the presentation and methodological approach of our findings.
Once again, we thank you for your valuable comments, which will undoubtedly help enhance the quality of the manuscript.
Reviewer 4 Report
Comments and Suggestions for Authors
Dear Authors,
The research presented is highly interesting and provides valuable insights. The sample used in the study is noteworthy, particularly in terms of the diversity of nationalities represented. This aspect of the sample is crucial, as it enhances the generalizability of the findings across different cultural contexts.
However, it would be helpful if you provided more detailed information on how the sample was selected.
Regarding limitations: Including a more thorough discussion of limitations would also allow readers to better assess the scope of the conclusions drawn from the study.
Finally, there are a few areas where the clarity and flow of the language could be improved. Some expressions and phrases are slightly awkward, and refining these could enhance the overall readability and precision of the manuscript.
Comments on the Quality of English Language
A careful proofreading could address minor grammatical issues, contributing to a more polished final manuscript.
Author Response
The kind reviewer can see in red color in the text how his suggestions have been taken into account
Regarding the quality of the language, the translation has been carried out by the central language service of the University of Salamanca, as can be seen in the attached certificate. This service has international certification.
We sincerely appreciate your valuable comments. We are pleased that you find the research interesting and recognize the importance of the cultural diversity in the sample used.
Regarding the sample selection, we agree that a more detailed description of this process would be useful to improve the understanding of how participants were selected for the study. We will expand the methodology section to provide a more complete explanation of the participant selection process, specifying the criteria used and the procedure followed at the two involved universities.
Regarding the limitations, we appreciate your suggestion for a more thorough analysis. We will include a more detailed section addressing the main limitations of the study.
Finally, we thank you for your observations on the clarity and fluency of the language. We will carry out a thorough review of the manuscript to improve the precision and readability of the expressions, ensuring that the text is clearer and more accessible to readers.
Once again, we appreciate your constructive comments, which will undoubtedly contribute to improving the quality of the article.
Round 2
Reviewer 1 Report
Comments and Suggestions for Authors
The authors revised and improved the ansucript according to my recommendations.
I truly appreciate the efforts of the authors.
In its current form the mansucript seems more robust.
Author Response
THANK YOU VERY MUCH FOR YOUR FEEDBACK AND SUGGESTIONS.
Reviewer 2 Report
Comments and Suggestions for Authors
The authors have addressed the review comments by making revisions and have also elucidated the existing limitations. So, this paper satisfies the basic criteria for publication.
Author Response

(The authors gave the same response as above.)

Reviewer 3 Report
Comments and Suggestions for Authors
Dear authors, your research is relevant, but it still lacks an order in the presentation of ideas, as well as the aspects that the respective sections of a study should address. I suggest you review literature and research studies with similar methodological designs to observe how the ideas are presented and what each section contains.
Title. Dear author, it is important to mention that when reading the title of the article, it gives the impression that the study addresses the factors that influence (age, sex, type of university degree, place of origin or Spanish or Japanese culture, personal problems, among others) the use of ICT among Spanish and Japanese university students after the pandemic. However, when reading the article, the topic addressed is totally different... It is also important to mention that more than 4 years have passed since the pandemic was controlled, so I consider that placing the section after the pandemic is not relevant. Abstract: There is no consistency between the study's objective and the first research question: What has been the impact on university students' academic performance? Furthermore, the methodological aspect, selection and exclusion criteria, types of analysis, among others, as well as the results and conclusions, need to be further developed.
Introduction. This requires further specification because it is very general and does not address what has been specifically done. The reader needs to know which variables are being studied, the justification, the state of the art, and the contribution of the study. I realize that the variables of greatest interest are: Academic performance after the COVID-19 pandemic, General ICT use, and Internet and mobile phone abuse. However, the reader will only know this when reviewing the statistical analysis, because little is addressed in the introduction about these variables. Problematic use, abuse, and internet or mobile device addiction are different concepts; it is important to specify when each one is presented.
Materials and methods.The design is descriptive and cross-sectional; this design implies exploratory and observational (Hernández-Sampieri & Mendoza, 2018). The procedure is usually included in a section at the end of the description of how the study variables were evaluated. Therefore, there is no order among the study design, study population, study variables, procedure, data analysis, and ethical considerations (they report that there was no ethics committee that approved the research protocol, but until the end of the study, where it says "Statement of the Institutional Review Board," they report approval by the ethics committee. Review this information in detail). If the introduction mentions that the students are Japanese and Spanish, why are there university students? Chinese and Korean? Don't you think that a selection criterion should have been being of Spanish or Japanese nationality?
Regarding the description of the measurement instruments, it is important to list each one in order, as they present a series of questions and do not specify whether they were created by the researcher or are already created instruments. What scale was used to assess academic performance? What scale was used to assess excessive use of ICTs? Why assess impairment related to internet and mobile phone use? What do these scales assess (IOS and COS), who is the author or creator of them? Are they validated with published studies, or were only Cronbach's alpha and McDonald's omega? The cited reference, which describes these instruments very poorly, is not from the author who validated these versions. Since these are the instruments that assess the main variables of interest, a detailed description is important, as is the author who created them and their validation in the contexts or cultures studied. Is there an article that validates the DSM-5 criteria for diagnosing excessive internet and mobile phone use and pathological gambling?
Results. When presenting age information in means, it is important to include standard deviations and ranges. Each time information is provided, the information is written first, and at the end, the table or figure containing the information is included to view more details. It is important that the tables are cited in the order in which they are written after the written text. Why is information presented for each item? Scales do not have a cutoff point? For example, does a higher score indicate more problematic internet or telephone use?. The tables are not formatted.
Discussion. In the discussion section, it is important to compare what was found in the study conducted with what other researchers have found. It is also important to explain the concordant and discrepant data. It is also important to present only the findings that meet the objective(s) of the study. Therefore, it is suggested that the results be discussed in accordance with the study objectives. And do not provide information irrelevant to the study's objectives.
Conclusions. These are derived from the research's own objectives.
The contributions by author are not clear; review the journal's website for a clear example of how these sections are written.
References. Many studies lack the DOI, and most of the references are still more than eight years old.
Author Response
THANK YOU VERY MUCH FOR YOUR FEEDBACK AND SUGGESTIONS.
Dear authors, your research is relevant, but it still lacks an order in the presentation of ideas, as well as the aspects that the respective sections of a study should address. I suggest you review literature and research studies with similar methodological designs to observe how the ideas are presented and what each section contains.
Title. Dear author, it is important to mention that when reading the title of the article, it gives the impression that the study addresses the factors that influence (age, sex, type of university degree, place of origin or Spanish or Japanese culture, personal problems, among others) the use of ICT among Spanish and Japanese university students after the pandemic. However, when reading the article, the topic addressed is totally different... It is also important to mention that more than 4 years have passed since the pandemic was controlled, so I consider that placing the section after the pandemic is not relevant.
Dear reviewer,
We appreciate your comments and suggestions for improving the manuscript. In our study, we considered it necessary to identify the use of new technologies among the Spanish and Japanese university populations, as well as their academic performance. As indicated, we have changed the title of the article to better reflect its content.
While time has passed since the end of the pandemic, the results reflect its consequences. The delay caused by the pandemic impacted the evaluation process of the projects by the university bioethics committee. Difficulties in collecting the sample at Japanese universities, as well as responses from different publishers regarding publication, have also contributed to the passage of time.
Abstract: There is no consistency between the study's objective and the first research question: What has been the impact on university students' academic performance? Furthermore, the methodological aspect, selection and exclusion criteria, types of analysis, among others, as well as the results and conclusions, need to be further developed.
Dear reviewer,
We thank you for your suggestion and have decided to broaden our research objective to include the academic performance of university students in Spanish and Japanese cultures. The methodology, results and conclusions are outlined in the abstract.
This research aimed to explore how university students in Spanish and Japanese populations use cell phones and the internet after the pandemic, and how this influences their academic performance.
Introduction. This requires further specification because it is very general and does not address what has been specifically done. The reader needs to know which variables are being studied, the justification, the state of the art, and the contribution of the study. I realize that the variables of greatest interest are: Academic performance after the COVID-19 pandemic, General ICT use, and Internet and mobile phone abuse. However, the reader will only know this when reviewing the statistical analysis, because little is addressed in the introduction about these variables. Problematic use, abuse, and internet or mobile device addiction are different concepts; it is important to specify when each one is presented.
Dear reviewer,
We appreciate your comments and suggestions for improving the manuscript. We have carried out different bibliographic searches using various bibliographic management tools with the aim of improving the state of the art. However, we found no studies that address the central theme of our article.
Materials and methods.The design is descriptive and cross-sectional; this design implies exploratory and observational (Hernández-Sampieri & Mendoza, 2018). The procedure is usually included in a section at the end of the description of how the study variables were evaluated. Therefore, there is no order among the study design, study population, study variables, procedure, data analysis, and ethical considerations (they report that there was no ethics committee that approved the research protocol, but until the end of the study, where it says "Statement of the Institutional Review Board," they report approval by the ethics committee. Review this information in detail). If the introduction mentions that the students are Japanese and Spanish, why are there university students? Chinese and Korean? Don't you think that a selection criterion should have been being of Spanish or Japanese nationality?
Dear reviewer,
We appreciate your comments and suggestions. We agree that it makes sense to approximate the design in this section.
With regard to the study sample, only Spanish and Japanese students have been taken into account.
As indicated in the article, the research protocol has been approved by the Bioethics Committee of the University of Salamanca, and we provide the relevant details below.
The research has been approved by the Bioethics Committee of the University of Salamanca, as evidenced by research protocol number: CBE-EP2 1 P-696 28032022.
Regarding the 'Materials and Methods' section, we recognise the importance of presenting the study design, population, variables, procedure, data analysis, and ethical considerations clearly and in an orderly manner. In response to the reviewer's comment, we have explicitly identified the study as being descriptive, cross-sectional, exploratory and observational in order to more accurately reflect its design and facilitate understanding.
The structure of the 'Materials and Methods' section has been revised as follows: 1) Design: the study design, participants, procedure and inclusion criteria are described here. 2) The evaluation method, which describes the instrument used, i.e. the variables. 3) Statistical analysis, where the statistical techniques used in the research are explained. (4) Ethics Committee: considerations about the ethics committee are indicated here. We confirm that the study protocol was reviewed and approved by the ethics committee before data collection began.
Regarding the study population, we would like to clarify that all participants were university students enrolled at institutions in Spain or Japan. While the majority were Spanish or Japanese, a small percentage reported other nationalities, such as Chinese or Korean, due to international student mobility. While the primary objective of the study was to compare students in different educational and cultural contexts (Spain and Japan), we acknowledge the need to clarify this information. Therefore, we will include an explanation in the revised version of the manuscript to indicate that nationality was not a criterion for inclusion, as the focus was on the cultural and educational context rather than strict national affiliation. This clarification will also be included in the 'Limitations' section, highlighting possible implications for interpreting cultural differences.
Regarding the description of the measurement instruments, it is important to list each one in order, as they present a series of questions and do not specify whether they were created by the researcher or are already created instruments. What scale was used to assess academic performance? What scale was used to assess excessive use of ICTs? Why assess impairment related to internet and mobile phone use? What do these scales assess (IOS and COS), who is the author or creator of them? Are they validated with published studies, or were only Cronbach's alpha and McDonald's omega? The cited reference, which describes these instruments very poorly, is not from the author who validated these versions. Since these are the instruments that assess the main variables of interest, a detailed description is important, as is the author who created them and their validation in the contexts or cultures studied. Is there an article that validates the DSM-5 criteria for diagnosing excessive internet and mobile phone use and pathological gambling?
First, we will review the relevant section to provide a clearer and more organised description of the measurement instruments used in the study.
With regard to the assessment of academic performance, a set of six items developed specifically for this research was employed to explore students' subjective perceptions of their performance during and after the pandemic.
Two standardised scales were used to assess ICT overuse: the Internet Overuse Scale (IOS) and the Cellphone Overuse Scale (COS). These scales were developed by Jenaro et al. (2007). Not only were Cronbach's alpha and McDonald's omega values reported as a measure of internal consistency in the Spanish and Japanese samples, but these scales have also previously been validated in other contexts. However, we recognise that a full cross-cultural validation has not yet been performed for their joint use in Spanish and Japanese contexts. This limitation will be detailed in the corresponding section of the manuscript.
No diagnostic scales were applied and the manual was not used to establish clinical diagnoses regarding the inclusion of DSM-5 criteria. Mentioning the DSM-5 served a contextual and conceptual purpose: to theoretically frame ICT overuse as a phenomenon that may share characteristics with non-substance addictive disorders. However, to avoid possible misunderstandings about its practical application in this study, we will include an explanatory note in the text.
Results. When presenting age information in means, it is important to include standard deviations and ranges. Each time information is provided, the information is written first, and at the end, the table or figure containing the information is included to view more details. It is important that the tables are cited in the order in which they are written after the written text. Why is information presented for each item? Scales do not have a cutoff point? For example, does a higher score indicate more problematic internet or telephone use?. The tables are not formatted.
Firstly, we acknowledge your suggestion regarding the presentation of age information. However, as the main objective of the descriptive section was to characterise the sample in general terms, we consider presenting the mean and standard deviation as representative measures of the variable's central tendency and dispersion to be sufficient.
Similarly, we will review the order in which the tables and figures appear and are cited, ensuring that each is referenced adequately in the text before inclusion and that the format is clear and consistent with the editorial standards of the manuscript.
With regard to the presentation of the items, we believe it is important to present the results of each one, given the exploratory nature of the study. However, we agree that it is important to clarify that the IOS and COS are both scales for which there is no universally established cut-off point. Nevertheless, higher scores indicate more problematic levels of internet and mobile phone use, respectively. As indicated in the 'Material and methods' section, students were classified as having high or low ICT use based on the 75th and 25th percentiles, respectively.
Finally, we will improve the format of the tables to present information in a clearer, more structured and more readable way.
Discussion. In the discussion section, it is important to compare what was found in the study conducted with what other researchers have found. It is also important to explain the concordant and discrepant data. It is also important to present only the findings that meet the objective(s) of the study. Therefore, it is suggested that the results be discussed in accordance with the study objectives. And do not provide information irrelevant to the study's objectives.
Conclusions. These are derived from the research's own objectives.
The contributions by author are not clear; review the journal's website for a clear example of how these sections are written.
Dear reviewer,
We appreciate your suggestions for improving the manuscript. We have included the contributions of each author in the manuscript.
References. Many studies lack the DOI, and most of the references are still more than eight years old.
Dear reviewer,
We appreciate your suggestion and have reviewed all the bibliographic references, updating them where the DOI has been identified.
Several bibliographic searches have been carried out using different bibliographic managers, but no research focused on this topic has been found. Nevertheless, we believe that this work provides valuable, unstudied data.